# Study on the Effects of a π Electron Conjugated Structure in Binuclear Metallophthalocyanines Graphene-Based Oxygen Reduction Reaction Catalysts

**DOI:** 10.3390/nano10050946

**Published:** 2020-05-15

**Authors:** Gai Zhang, Bulei Liu, Yufan Zhang, Tiantian Li, Weixing Chen, Weifeng Zhao

**Affiliations:** 1School of Materials Science and Chemical Engineering, Xi’an Technological University, Xi’an 710021, China; gaomin@st.xatu.edu.cn (B.L.); zhangyufan@st.xatu.edu.cn (Y.Z.); chenwx@xatu.edu.cn (W.C.); 2Key Laboratory of Synthetic and Natural Functional Molecule Chemistry of Ministry of Education, College of Chemistry & Material Chemistry, Northwest University, Xi’an 710069, China; 2015116027@stumail.nwu.edu.cn

**Keywords:** metallophthalocyanines, PSS-Graphene, π-π interactions, ORR

## Abstract

The high overpotentials for oxygen reduction reaction (ORR) create an extremely negative impact on the energy efficiency of the air-based battery systems. To overcome this problem, binuclear ball-type metallophthalocyanines containing methoxy substituents (M_2_Pc_2_(EP)_4_, M = Fe(II), Co(II) and Zn(II)) were wrapped with polystyrene sodium sulfonate (PSS) modified graphene oxide (GO), using a facilely “solvothermal π-π assembly” method to prepare M_2_Pc_2_(EP)_4_/PSS-Gr composites. Compared with the commercial Pt/C catalysts, the M_2_Pc_2_(EP)_4_/PSS-Gr composites enhanced the catalytic activity of oxygen reduction reaction. The π electron conjugated structure of the MN_4_-type phthalocyanine macrocyclic system strongly influenced the one-step four-electron electrocatalytic process of the M_2_Pc_2_(EP)_4_/PSS-Gr composites. Moreover, the π-π interactions between the M_2_Pc_2_(EP)_4_ and PSS-Gr dramatically enhanced the π electron density in the conjugated structure and oxygen could be reduced more easily. The electrocatalytic activity test was displayed in the order of Fe_2_Pc_2_(FP)_4_/PSS-Gr > Co_2_Pc_2_(EP)_4_/PSS-Gr > Zn_2_Pc_2_(EP)_4_/PSS-Gr. The results indicated that the catalytic performance of M_2_Pc_2_R_n_ could be enhanced by the modification of π electron conjugated structure of M_2_Pc_2_(EP)_4_ and carbon materials.

## 1. Introduction

With the exhausting of traditional energy sources and increasing concerns of environmental pollution, it is urgent to explore and utilize clean energy. Fuel cells have become the research hotspots for new energy development because of their low cost, simple structure, special energy density and other merits [1,2,3]. Nevertheless, the high overpotentials for the oxygen reduction reactions create an extremely negative impact on the energy efficiency of air-based battery systems [4]. Although Pt and its alloys contribute significantly in decreasing the overpotential of oxygen reduction reaction (ORR) because of their high catalytic activity, they are limited in the scarcity of reserves, high price and lack of excellent methanol resistance [5]. To overcome this problem, extensive efforts have been implemented to study low-cost, non-precious metal substitute catalysts for Pt/C catalysts [6,7]. Specially, many reports have pointed out that the ability to carry oxygen molecules plays a key role in the ORR process [8,9]. Among the large variety of metal macrocycles, metal-N_4_-chelates, like metalloporphyrin (MPs) and metallophthalocyanine (MPcs), which have an 18 π electrons conjugated structure, have the biomimetic oxygen carrier functions similar to the naturally occurring hemeproteins [10]. Due to the delocalization effect and weakly bonding character of π electron clouds in a conjugated structure, the metal-N_4_-chelates with π electron conjugated structure could be oxidized and reduced much more easily, establishing them as a promising precursor for preparing ORR catalysts [11,12,13]. Compared with MPs, the periphery of the benzene ring of phthalocyanines can be modified with a variety of substituents, giving MPcs a variety of chemical structures and properties [14,15,16]. The chemical natures of the ligand and the central ion strongly influence the catalytic properties of the metal-N_4_-chelates macrocycles. In the ORR process, O_2_ is coordinated to the M of MPcs and the electrons then migrate from the MPcs to O_2_ to form adduct MPc-O_2_^−^ [17,18]. Considering a high π electron density of the conjugated structure, metallophthalocyanines have an excellent catalytic activity for oxygen reduction reaction [19,20,21].

In recent years, a new class of ball-type M_2_Pc_2_ compounds containing two metal centrals has drawn much attention because of its attractive structure [22,23]. Both the face-to-face monomer distance and the d filling of the metal central ions can significantly affect the ball-type Pcs’ physical and chemical behavior [24,25]. For the M_2_Pc_2_ with a strong ligand-field effects of MN_4_-chelates, the catalytic activity is mainly affected by the chemical structure of the phthalocyanines, the radius and the d-filling of the metal central ions [26]. In addition, the presence of bridging units including electronegative groups highly enhanced catalytic activity of the phthalocyanines. The multiple electron-withdrawing substituents on the periphery of metal Pcs increased their catalytic activity [25]. The results pointed out that the possibility of the increase of the catalytic performance by the modification of the main M_2_Pc_2_ skeleton, and encouraged us to design phthalocyanine compound containing methoxy substituents. Furthermore, reports have also pointed out that MPc loaded on carbon materials is conducive to enhance the catalytic activity for the oxygen reduction reaction [27,28]. Graphene oxide has a high conductivity, excellent electron mobility and relatively high theoretical specific surface areas, establishing itself as a good candidate to provide a pathway for fast electron transferring and to prevent the aggregation of MPcs nanoparticles [29,30]. It has demonstrated that the catalytic activity of the π electron conjugated structure towards oxygen reduction reaction are related to the strong π-π supramolecular interaction between MPc compounds and graphene, which promote electron transfer between them leading to an apparent improvement for the oxidative reactions [31,32,33,34]. However, pristine GO possesses few proton-conducting groups, which negatively enhances the conductivity of GO. Therefore, grafting GO with a sulfonated group (-SO_3_H; PSS-Gr) using various methods has been studied, as this strategy not only improved the conductivity of the GO, but also enabled the better dispersibility of GO [35].

Some studies have provided valuable mechanistic insights for the formation of M_2_PcR_n_-O_2_ intermediates. However, the study on the effects of the π electron conjugated structure and the π-π interactions between the M_2_Pc_2_(EP)_4_ and PSS-Gr generally restrict the design of the M_2_Pc_2_R_n_-based catalysts. In this work, composites of binuclear ball-type metallophthalocyanines with methoxy substituents (M_2_Pc_2_(EP)_4_, M = Fe(II), Co(II) and Zn(II)) were loaded on the surface of PSS-Graphene (PSS-Gr), to enhance the catalytic activity and stability for ORR based on the π electron conjugated structure of the MN_4_-type phthalocyanine macrocyclic system, and the π-π supramolecular interaction between MPc nanoparticles and graphene.

## 2. Experimental

### 2.1. Materials

4, 4-bis (4-hydroxyphenyl) pentanoic acid (98%) and 4-nitrophthalonitrile (99%) were purchased from Aladdin (Shanghai, China). The other chemicals were of analytical grade and were used without further purification. The target M_2_Pc_2_(EP)_4_ compounds were prepared by taking bisphthalonitrile and the corresponding metal salts (M = Fe(II)*,* Co(II), and Zn(II)) as the raw materials in the dimethylaminoethanol (DMAE). This was done according to the literature [25].

Preperation of M_2_(II)Pc_2_(EP)_4_: Bisphthalonitrile (0.56 g, 1.02 mmol), Zn(OAc)_2_ 2H_2_O (0.112 g, 0.52 mmol) and 4 mL DMAE were poured into a Teflon-lined autoclave at 220 °C for 4 h (Figure 1). The reaction mixture was then poured into methanol to produce a precipitate, and the precipitate was washed sequentially with acetic acid, water and methanol. The crude product was then dissolved in DMF and reprecipitated by gradually adding methanol to the solution. The precipitate was washed again as in the previous method followed by centrifugation and dried at 100 °C in an oven.

*[2′,10′,16′,24′-{Tetrakis-4,4′-bis (4- (3-cyano-4-isocyanophenoxy) phenyl) pentanoate diphthalocyaninato) dizinc (II)]**:* Zn_2_(II)Pc_2_(EP)_4_ (0.3815 g, yield 64.21%), green solid, m.p. > 300 °C. IR (KBr) ν_max_/cm^−1^: 1715 (ν_C =_ o); 1381 (ν_C =_
_C_); 892 (ν_M-N_); 2972, 1009 (ν_C__-H(Pc)_), 1229 (ν_Ar-__O-Ar_), 740 (ν_Pc_). UV-Vis (DMF) λ_max_/nm: B band: 267, 354; Q band: 612, 677. Anal. Cald. for C_32_H_16_N_8_Zn: C, 69.78; H, 4.10; N, 9.58; Found: C, 69.21; H, 4.54; N, 9.29.

*[2′,10′,16′,24′-{Tetrakis-4, 4′-bis (4- (3-cyano-4-isocyanophenoxy) phenyl) pentanoate diphthalocyaninato) diiron (II)]**:* Fe_2_(II)Pc_2_(EP)_4_ (0.2120 g, yield 36.64%)**,** olive green solid, m.p. > 300 °C. IR (KBr) ν_max_/cm^−1^: 1713 (ν_C =_ o); 1401 (ν_C =_
_C_); 874 (ν_M-N_); 2970, 1011 (ν_C__-H(Pc)_), 1234 (ν_Ar-__O-Ar_), 752 (ν_Pc_). UV-Vis (DMF) λ_max_/nm: B band: 267, 323; Q band: 621, 668. Anal. Cald. for C_32_H_16_N_8_Fe: C, 70.07; H, 4.12; N, 9.62; Found: C, 69.68; H, 3.63; N, 9.27.

*[2**′,10**′,16**′,24**′-{Tetrakis-4, 4**′-bis (4- (3-cyano-4-isocyanophenoxy) phenyl) pentanoate diphthalocyaninato) dicobalt (II]*: Co_2_(II)Pc_2_(EP)_4_(0.3021 g, yield 51.52%), dark blue solid, m.p. > 300 °C. IR (KBr) ν_max_/cm^−1^: 1707 (ν_C =_ o); 1387 (ν_C =_
_C_); 891 (ν_M-N_); 2970, 1011 (ν_C__-H(Pc)_), 1231 (ν_Ar-__O-Ar_), 745 (ν_Pc_). UV-Vis (DMF) λ_max_/nm: B band: 268, 362; Q band: 668. Anal. Cald. for C_32_H_16_N_8_Co: C, 69.97; H, 4.11; N, 9.61; Found: C, 69.55; H, 4.56; N, 9.19.

### 2.2. Characterization

Transmission electron microscopy (TEM) analysis was performed on a JEOL JEM-2010 electron microscope at 200 kV. The system consists of a vertical SEM column that was situated at 36° relative to the FIB column, operating at 3 kV to optimize surface sensitivity. The IR spectra were recorded on a Germany Bruker Vertex70 spectrometer (Bruker, Karlsruhe, Germany). The UV-vis absorbance was recorded on a UV/visible spectrophotometer (UV-1600, Shanghai, China) using a quartz cell with a path length of 10 mm at room temperature. X-ray photoelectron spectroscopy (XPS) was measured using an Axis Ultra spectrometer with an Al (Mono) Kα X-ray source (1486.6 eV). The binding energies (BE) were normalized to the signal for adventitious carbon at 284.8 eV. The electrocatalytic performance was evaluated by an electrochemical workstation CHI 660E (Shanghai CHENHUA Company, Shanghai, China) and a Pine Instrument Company AF-MSRCE modulator rate rotator (Grove, PA, USA) in a 0.1 M KOH solution, separately.

### 2.3. Synthesis

#### 2.3.1. Polystyrene Sodium Sulfonate Modified Graphene (PSS-Gr) Preparation

Graphene oxide (GO), obtained from Aladain Co. Ltd. (Shanghai, China) was dispersed in 100 mL distilled water with an ultrasonic technique, followed by the addition of 1 g of sodium polystyrene sulfonate at room temperature, and stirred for 12 h. Then, 2 mL of hydrazine hydrate was then added. The crude products were washed with deionized water, ethanol and n-pentanol, followed by centrifugation and then drying at 100 °C.

#### 2.3.2. Preparation of Fe_2_Pc_2_(EP)_4_/PSS-Gr

The Fe_2_Pc_2_(EP)_4_/PSS-Gr composites were synthesized by a facile “solvothermal π-π assembly” method, using PSS-Gr and Fe_2_Pc_2_(EP)_4_ as the precursors (Figure 1). In brief, 0.1503 g of PSS-Gr powder was dispersed in 10 mL of DMF solution with an ultrasonic technique for 10 min, followed by the addition of 0.1100 g metallophthalocyanine Fe_2_Pc_2_(EP)_4_. The resulting solution was ultrasonically dispersed for 2 h. Nitrogen gas was then bubbled into the mixture to remove oxygen and was then poured into a Teflon-lined autoclave at 160 °C for 24 h. All of the crude products were washed with DMF, deionized water, ethanol and n-pentanol, followed by centrifugation and drying at 100 °C in an oven.

The Co_2_Pc_2_(EP)_4/_PSS-Gr and Zn_2_Pc_2_(EP)_4_/PSS-Gr composites were prepared under the same conditions.

### 2.4. Evaluation of the Electrocatalytic Activity

The electrocatalytic performance of M_2_Pc_2_(EP)_4_/PSS-Gr composites for the oxygen reduction reaction was measured by cyclic voltammetry (CV) and rotating disk electrode (RDE) techniques in a 0.1 M NaOH solution at room temperature. Specifically, the modified glassy carbon electrode is used as a working electrode. The reference electrode is saturated calomel electrode (SCE) and the platinum (Pt) wire electrode is used as counter electrode, respectively. The cyclic voltammetry tests were investigated in an O_2_-saturated 0.1 M NaOH solution with the scan rate of 100mV s^−1^. The rotating disk electrode (RDE) test was measured with a glassy carbon electrode (5 mm diameter) in O_2_-saturated 0.1 M NaOH solution under quasistationary conditions (5 mV·s^−1^ sweep rate) at 25 °C. The rotating disk electrode (RDE) was performed on a CHI 660E electrochemical workstation with an AF-MSRCE modulator rate rotator (Pine Instrument Company) using a standard three-electrode system. A platinum ring electrode and glassy carbon disk (5.61 mm diameter) was selected as the working electrode. The collection efficiency of the platinum ring was 37%.

## 3. Results and Discussion

### 3.1. Morphology and Structure of M_2_Pc_2_(EP)_4_/PSS-Gr Composites

The surface morphology of the M_2_Pc_2_(EP)_4_/PSS-Gr composites was studied by SEM and TEM images. As illustrated in Figure 2, a wrinkled paper-like feature was observed for the PSS-Gr, indicating a typical feature of the graphene sheet. In contrast to the TEM micrograph of PSS-Gr that appears transparent (Figure 2b), numerous dark particles of M_2_Pc_2_(EP)_4_, as indicated by the white arrow, were observed on the PSS-Gr layer from the TEM image of the M_2_Pc_2_(EP)_4_/PSS-Gr composites. It can be seen that M_2_Pc_2_(EP)_4_ nanoparticles were dispersed uniformly on the PSS-Gr surface.

The UV-Vis spectra of the Fe_2_Pc_2_(EP)_4_/PSS-Gr composites is shown in Figure 3. The “Q-band” of Fe_2_Pc_2_(EP)_4_ appeared at around 668 nm, because of the π-π originating from the HOMO (a_1u_ and a_2u_) to the LUMO orbitals (e_g_) [25]. The absorption at 621 nm is attributed to the dimmers of Fe_2_Pc_2_(EP)_4_. The spectrum of the PSS-Gr did not exhibit any obvious absorption peaks from 300 to 900 nm, while that of the Fe_2_Pc_2_(EP)_4_/PSS-Gr composites in DMF showed an absorption peak at 741 nm. Compared with the Fe_2_Pc_2_(EP)_4_, the absorption of the Fe_2_Pc_2_(EP)_4_/PSS-Gr samples are red shifted from 668 nm to 741 nm, and the absorption of Fe_2_Pc_2_(EP)_4_ dimmers at 621 nm disappeared. This suggests a strong π-π supramolecular interaction between the Fe_2_Pc_2_(EP)_4_ and PSS-GR [36,37,38], which will prevent the aggregation of Fe_2_Pc_2_(EP)_4_ compounds and enhance the π electron density in conjugated structure. Moreover, the strong π-π supramolecular interaction facilitates electron transfer between them, leading to the observed improvement for the oxygen reduction reaction.

To further illustrate the composition and chemical status of the as-prepared catalysts, the Fe_2_Pc_2_(EP)_4_/PSS-Gr were also analyzed by XPS. Figure 4a displays the XPS survey spectra of GO, PSS-Gr, Fe_2_Pc_2_(EP)_4_, and Fe_2_Pc_2_(EP)_4_/PSS-Gr composites. The results showed that the Fe_2_Pc_2_(EP)_4_/PSS-Gr catalysts are composed of O, C, N and Fe elements, which further confirms the existence of Fe_2_Pc_2_(EP)_4_ in the as-prepared samples. Compared with GO, the obviously decreasing O content for PSS-Gr indicates that a relatively high degree of reduction had been achieved during the hydrothermal process. The high-resolution N1s spectrum of the composite Fe_2_Pc_2_(EP)_4_ and Fe_2_Pc_2_(EP)_4_/PSS-Gr are shown in Figure 4c,d. The two asymmetric broad peaks of N1s for Fe_2_Pc_2_(EP)_4_ are located at 398.1 eV and 399.4 eV, which correspond to the signals of C-N and C = N of pyrrolic ring in the phthalocyanine macrocycle, respectively [39,40]. What is more interesting is that the binding energy values of N1s in the Fe_2_Pc_2_(EP)_4_/PSS-Gr composites are obviously higher than those of pure Fe_2_Pc_2_(EP)_4_, which confirms a strong interaction between Fe_2_Pc_2_(EP)_4_ and PSS-Gr.

The high-resolution Fe2p spectrum of the catalysts is shown in Figure 4b. The peaks of Fe2p for Fe_2_Pc_2_(EP)_4_ are located at 723.9 eV and 710.6 eV, which correspond to the signals of Fe2p_1/2_ and Fe2p_3/2_ in the bivalent oxidation state, respectively. However, the binding energy values of Fe2p in the composites of Fe_2_Pc_2_(EP)_4_/PSS-Gr are obviously lower than those of pure Fe_2_Pc_2_(EP)_4_ [19]. The shift again confirms a π-π supramolecular interaction between Fe_2_Pc_2_(EP)_4_ and PSS-Gr, indicating the as-prepared composites of Fe_2_Pc_2(_EP)_4_ and PSS-Gr tend to form a hetero structure, rather than a physical mixture.

### 3.2. Effects of π Electron Conjugated Structure for ORR

The electrocatalytic activity of the M_2_Pc_2_(EP)_4_/PSS-Gr composites for ORR were first tested by the technique of CV. As shown in Figure 5, the ORR peak currents in the voltammograms showed that M_2_Pc_2_(EP)_4_/PSS-Gr composites interact well with O_2_ through the redox-active character of the M^2+^ cores. The π electron conjugated structure of the MN_4_-type phthalocyanine macrocyclic system strongly influences the redox-active character of the M^2+^ cores. Therefore, because of the d filling t_2g_^6^ e_g_^0^ of Fe(II) with a fully filled t_2g_ and unoccupied e_g_ orbitals, the central metal Fe(II) was favored to coordinate with dioxygen, and was oxidized to Fe(III) much more easily in the redox process [17]. The results showed that the Fe(II)/Fe(III) reduction peak current of O_2_ commenced at lower positive potentials (−0.15 V vs. SCE) than that of the Pt/C (−0.17 V vs. SCE). When the Fe(II)/Fe(III) redox transition occurs at low positive potentials, highly acidic Fe(III) species well interact well with dioxygen and provide a relatively low overpotential for ORR. Moreover, the reduction potential of Fe(III)/Fe(II) (−0.15 V vs. SCE) is more positive than that of Co(III)/Co(II) (−0.21 V vs. SCE) and Zn(III)/Zn(II)(−0.59V vs. SCE). The results indicated that the electrocatalytic activity is closely related to their radius and d filling of the active center ions. Zn_2_Pc_2_(EP)_4_/PSS-Gr shows a relatively low catalytic activity for ORR because of the fully filled t_2g_^6^ and e_g_^4^ orbitals, which make Zn(II) difficult to be oxidized to Zn(III) by the O_2_ in the vertical direction of the conjugate plane in a strong ligand-field. Whereas the ORR process occurs only on the phthalocyanine ring, and is independent of the central metal Zn(II).

In order to further investigate the influence of the catalysts structures for ORR, the linear sweep voltammetry (LSV) measurement was performed on a rotating disk electrode (RDE) at a scanning rate of 5 mV^−1^ in O_2_-saturated 0.1 M KOH solution with the rotational speed from 400 rpm to 2500 rpm. A comparison of the LSV results was recorded at 1600 rpm rotational speed using Pt/C, PSS-Gr, Fe_2_Pc_2_(EP)_4_ and Fe_2_Pc_2_(EP)_4_/PSS-Gr modified electrodes individually. The onset potential (E_0_) and the limiting diffusion current density (J_l_) were taken as measures of the catalytic activity. Compared with the ORR potentials of PSS-Gr (−0.17 V vs. SCE) and Fe_2_Pc_2_(EP)_4_ (−0.27 V vs. SCE), the onset potential of Fe_2_Pc_2_(EP)_4_/Gr was positively shifted to −0.09 V, which is close to the onset potential of Pt/C (−0.07 V). The results indicated that the electrocatalytic activity of Fe_2_Pc_2_(EP)_4/_PSS-Gr was enhanced by the π-π supramolecular interaction between Fe_2_Pc_2_(EP)_4_ compounds and PSS-Gr, which enhance the π electron density in Pc_2_(EP)_4_ conjugated structure, leading to an observed improvement for the oxygen reduction reaction [41,42].

Furthermore, it is well known that a one-step four-electron process has more excellent electrocatalytic performances than the two-electron process for an oxygen reduction reaction [43,44,45]. The total electron transfer number (n) in the ORR reaction is calculated by the Koutecky-Levich (K-L) equation given below (1) and (2):(1)$$\frac{1}{J}=\frac{1}{JK}+\frac{1}{JL}=\frac{1}{nFKC_{o}}+\frac{1}{B\omega^{1/2}},$$
(2)$$B=0.62nFC_{o}{\left(D_{0}\right)}^{2/3}\nu^{-1/6},$$
where J (mA/cm^2^) is the measured current density; J_K_ and J_L_ (mA/cm^2^) are the kinetic and diffusion-controlled current density, respectively; ω is the angular velocity of the rotating disk (ω = 2πN, N is the linear rotation speed in rpm); n is the total number of electron transferred per oxygen molecule in the ORR reaction; F is the Faraday constant; C_0_ is the bulk oxygen concentration; D_0_ is the diffusion coefficient of oxygen; and ν is the kinematic viscosity of the electrolyte.

The K-L points (J^−1^ vs. ω^−1/2^) of M_2_Pc_2_(EP)_4_/PSS-Gr at different voltages an exhibited excellent linearity and the slope is consistent (Figure 6). The linearity of the K-L plots and the near parallelism of the fitting lines suggested first-order reaction kinetics for ORR. The electron transfer number n is further calculated by the Koutecky-Levich (K-L) equation. The n for Fe_2_Pc_2_(EP)_4_/PSS-Gr was 3.89 at the voltage range from −0.4 V to −0.7 V, and was similar in the redox process at different potentials. The results indicated that the ORR mainly proceeded a one-step four electron process for Fe_2_Pc_2_(EP)_4_/PSS-Gr which is similar to the Pt/C catalysts.

The cyclic voltammetry curves of Pt/C and Fe_2_Pc_2_(EP)_4_/PSS-Gr were further investigated, revealing reveal the cross-effect in an O_2_ saturated 0.1 M NaOH solution containing 3 M CH_3_OH. As shown in Figure 7, the oxygen reduction peak of Pt/C catalyst at −0.20 V decreased significantly for the oxygen reduction reaction after adding methanol into the system, and a methanol oxidation peak appeared at 0.0 V. However, the peak current density and peak potential had no significant difference for the Fe_2_Pc_2_(EP)_4_/PSS-Gr catalyst under the same conditions. The results indicate that the Fe_2_Pc_2_(EP)_4_/Gr catalyst had a good methanol-tolerant performance in an alkaline medium in the presence of methanol.

A mechanism was proposed for ORR catalyzed by Fe_2_Pc_2_(EP)_4_/PSS-Gr based on the experiment results and references (Figure 8). The electron transfer number n for Fe(II)_2_Pc_2_(EP)_4_/PSS-Gr was 3.89 which was calculated by the Koutecky-Levich (K-L) equation. The results indicated that ORR mainly underwent a direct four electron pathway to result in OH^−^ production. First, the O_2_ molecule was bonded to the central metal ion Fe(II) (d-filling of t_2g_^6^ e_g_^0^) of the MN_4_-type Fe(II)_2_Pc_2_(EP)_4_ macrocyclic system by bridge adsorption [16,17]. Bridge adsorption can coordinate one oxygen molecule with two central metal ions Fe(II), so it can form more stable peroxide intermediates, which are more conducive to oxygen-oxygen bond (O-O) breaking [25,46,47]. The electrons on the π electron conjugated structure Pc_2_(EP)_4_ migrate to the central metal ion Fe(II) and then to the O_2_ molecule, to form an adduct Fe(III)_2_Pc_2_(EP)_4_·O_2_^−^, which showed the structure of Fe(III)-O-O-Fe(III) peroxo-species. The breaking of the O-O linkage for Fe(III)-O-O-Fe(III) peroxo species is expected to facilitate in the formation of OH^−^ ions. The electrons on PSS-Gr then migrated to Fe(III)_2_Pc_2_(EP)_4_ and Fe(III)_2_Pc_2_(EP)_4_ was reduced to Fe(II)_2_Pc_2_(EP)_4_ [19,32,33]. Therefore, the π electron conjugated structure of the MN_4_-type phthalocyanine macrocyclic system strongly influenced the one-step four-electron electrocatalytic process for ORR, and it effectively reduced the overpotential of the oxide reduction reaction [25]. Moreover, the performance of M_2_Pc_2_(EP)_4_ catalysts was lower than that of the metallophthalocyanines with trifluoro methyl linkages [25]. The linkage of highly electrophilic groups, like trifluoro methyl groups, to the macrocycles dramatically enhanced the electrocatalytic performance. The comparison of the performances of M_2_Pc_2_(EP)_4_ catalysts used in this work and in the reported article also points out the importance of the bridging units in the ball-type structure. More significantly, PSS-Gr dramatically enhanced the electrocatalytic activity of M_2_Pc_2_(EP)_4._ PSS-Gr provide pathway for fast electron transferring and to prevent the aggregation of M_2_Pc_2_(EP)_4_ catalysts. M_2_Pc_2_(EP)_4_ were loaded on the surface of PSS-Gr to enhance the catalytic activity and stability for the ORR, based on the π-π supramolecular interaction between MPcs molecules and graphene. The results indicated that PSS-Gr enhanced the catalytic activity and stability of M_2_Pc_2_(EP)_4_ for the ORR based on the π-π supramolecular interaction.

## 4. Conclusions

Composites of PSS-graphene-wrapped binuclear ball-type metallophthalocyanines with methoxy substituents (M_2_Pc_2_(EP)_4_, M = Fe(II), Co(II) and Zn(II)) were synthesized to enhanced the electrocatalytic activity for the oxygen reduction reaction of M_2_Pc_2_(EP)_4._ Compared to the commercial Pt/C catalysts, the M_2_Pc_2_(EP)_4_/PSS-Gr composites had a high electrocatalytic activity. The π electron conjugated structure of the MN_4_-type phthalocyanine macrocyclic system strongly influenced the four-electron electrocatalytic process. PSS-Gr enhance the catalytic activity and stability of the M_2_Pc_2_(EP)_4_ composites, based on the π-π supramolecular interaction between MPcs molecules and graphene. PSS-Gr provide a pathway for fast electron transferring and prevent the aggregation of M_2_Pc_2_(EP)_4_ catalysts. The results indicated that the catalytic performance of M_2_Pc_2_R_n_ could be enhanced by the modification of the π electron conjugated structure of M_2_Pc_2_(EP)_4_ and carbon materials.

## Figures and Tables

**Figure 1 nanomaterials-10-00946-f001:**
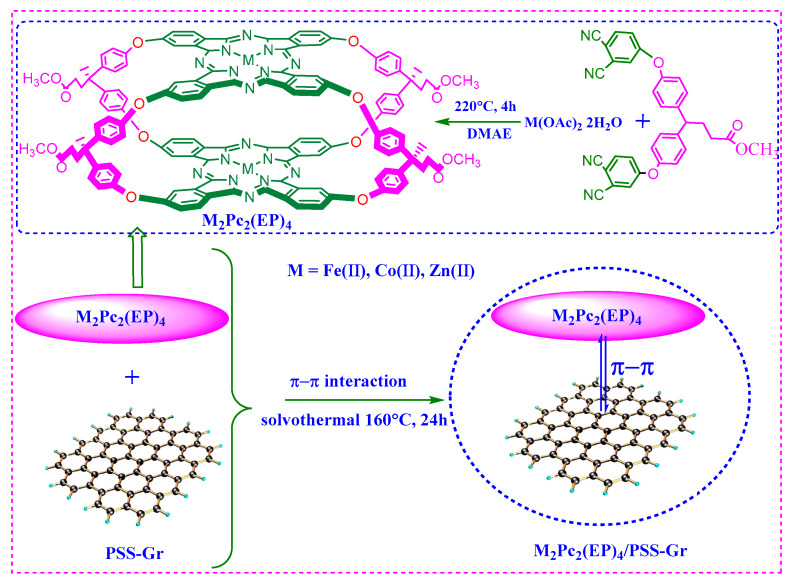
Schematic illustration of the preparation process for the M_2_Pc_2_(EP)_4_/polystyrene sodium sulfonate modified graphene (PSS-Gr) composites.

**Figure 2 nanomaterials-10-00946-f002:**
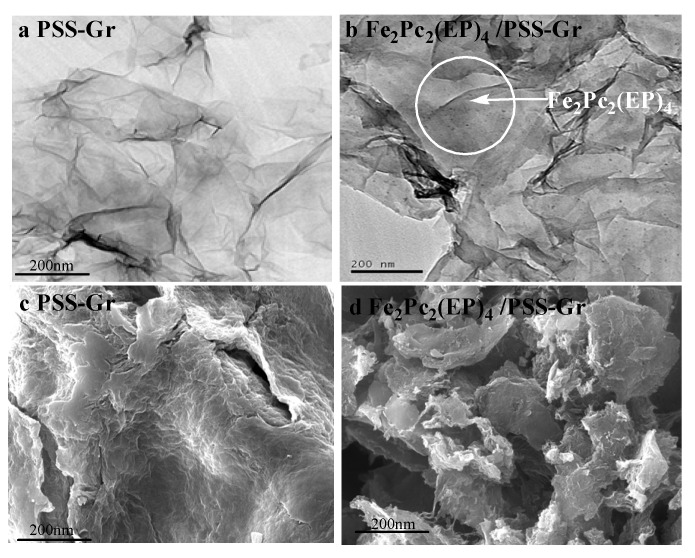
TEM and SEM images of PSS-Gr and Fe_2_Pc_2_(EP)_4_/PSS-Gr; (**a**), (**c**) TEM and SEM images of PSS-Gr; (**b**), (**d**) TEM and SEM images of Fe_2_Pc_2_(EP)_4_/PSS-Gr.

**Figure 3 nanomaterials-10-00946-f003:**
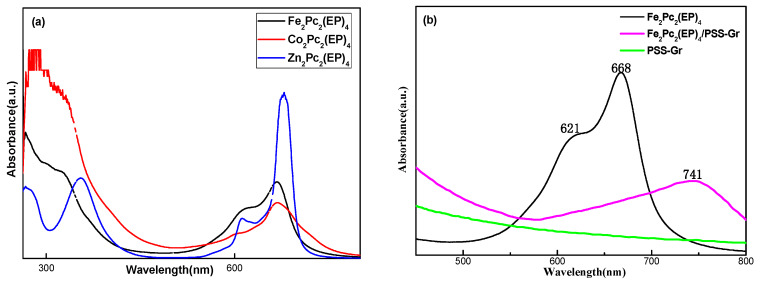
(**a**) UV-Vis spectra of Fe_2_Pc_2_(EP)_4_, Co_2_Pc_2_(EP)_4_, and Zn_2_Pc_2_(EP)_4_, and (**b**) UV-vis spectra of Fe_2_Pc_2_(EP)_4_/PSS-Gr, Fe_2_Pc_2_(EP)_4_ and PSS-Gr.

**Figure 4 nanomaterials-10-00946-f004:**
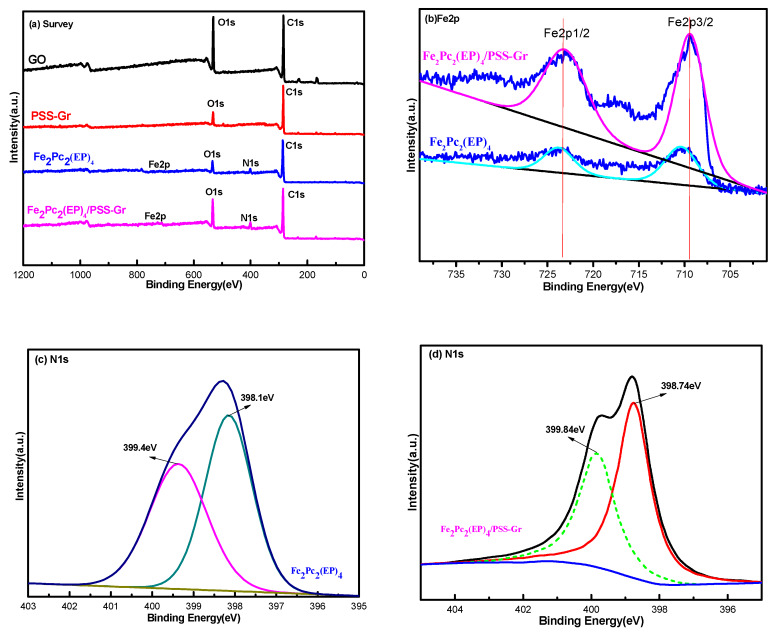
X-ray photoelectron spectroscopy (XPS) spectra of Fe_2_Pc_2_(EP)_4_/PSS-Gr; (**a**) XPS survey spectra of GO, PSS-Gr, Fe_2_Pc_2_(EP)_4_ and Fe_2_Pc_2_(EP)_4_/PSS-GR; (**b**), (**c**), (**d**) High resolution of Fe2p, N1s XPS spectra of Fe_2_Pc_2_(EP)_4_ and Fe_2_Pc_2_(EP)_4_/PSS-Gr.

**Figure 5 nanomaterials-10-00946-f005:**
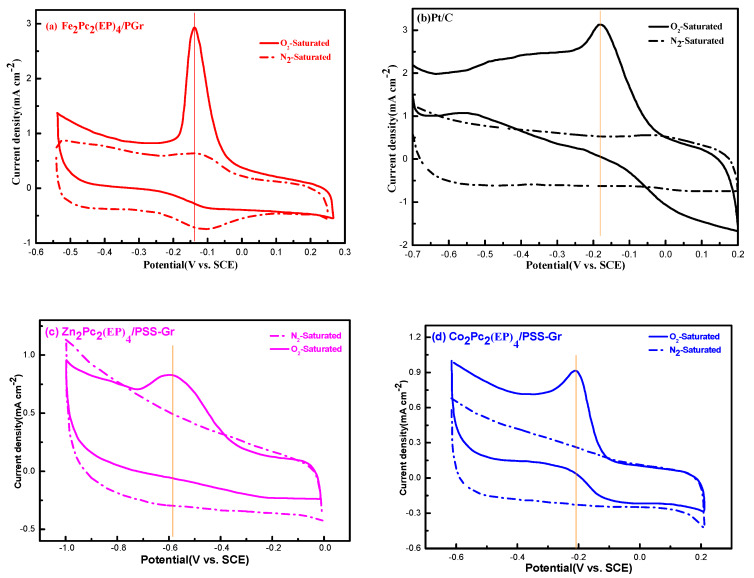
Cyclic Voltammograms curve of M_2_Pc_2_(EP)_4_/PSS-Gr and Pt/C catalysts on glassy carbon electrodes; (**a**), (**b**), (**c**), (**d**) CV curve of Fe_2_Pc_2_(EP)_4_/PSS-Gr, Pt/C, Zn_2_Pc_2_(EP)_4_/PSS-Gr, and Fe_2_Pc_2_(EP)_4_/PSS-Gr;

**Figure 6 nanomaterials-10-00946-f006:**
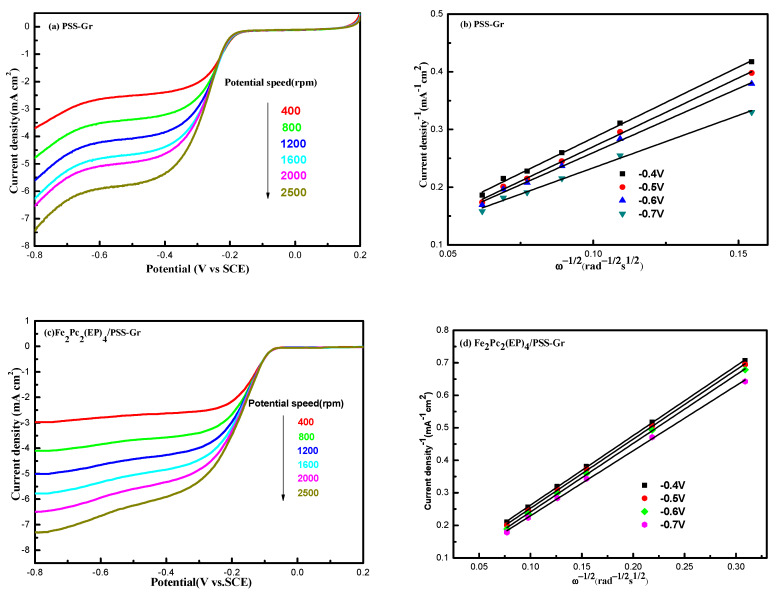
Electrocatalytic activities of Pt/C and Fe_2_Pc_2_(EP)_4_/PSS-Gr at different rotation speed in O_2_-saturated 0.1 M KOH; (**a**), (**c**), (**e**), (**g**) the linear sweep voltammetry (LSV) curve; (**b**), (**d**), (**f**) Koutecky-Levich (K-L) plots; (**h**) the number of electrons transferred.

**Figure 7 nanomaterials-10-00946-f007:**
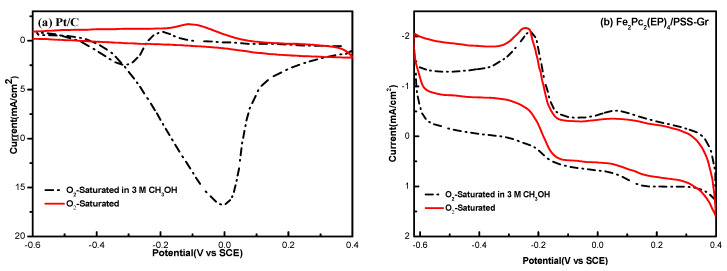
Cyclic voltammetry **(**CV) curves of Fe_2_Pc_2_(EP)_4_/PSS-Gr in O_2_-saturated in a 0.1 M KOH solution and O_2_-saturated 0.1 M KOH solution with 3 M methanol. (**a**) Pt/C, and (**b**) Fe_2_Pc_2_(EP)_4_/Gr.

**Figure 8 nanomaterials-10-00946-f008:**
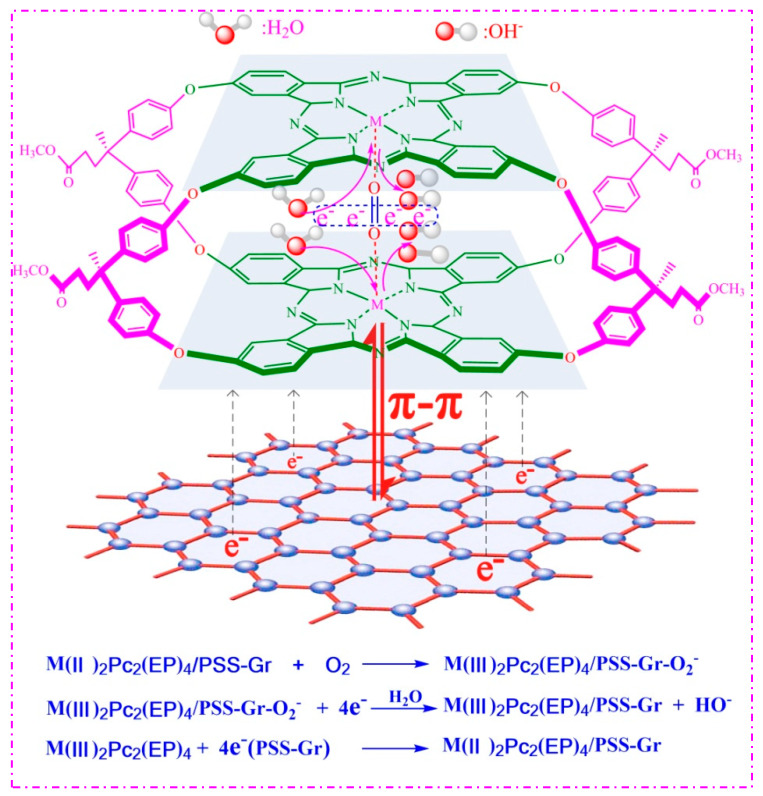
Mechanism for the oxygen reduction reaction (ORR) catalyzed by Fe_2_Pc_2_(EP)_4_/PSS-Gr.

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
