# Peer review of "Study on the Effects of a π Electron Conjugated Structure in Binuclear Metallophthalocyanines Graphene-Based Oxygen Reduction Reaction Catalysts"

_nanomaterials, 2020, doi:10.3390/nano10050946_

Round 1

Reviewer 1 Report

Recommendation: The manuscript is publishable after a major revision.

            The manuscript by G. Zhang and W. Zhao and co-workers deals with the development of catalysts for oxygen reduction reaction. Novel binuclear metallophthalocyanines were prepared and deposited on GO. Then the composite materials were tested as catalysts in oxygen reduction reaction. Composite materials based on binuclear metallophthalocyanines were already studied in this reaction. In this work, the efficiency of their deposition on GO surface is explored. The work is carefully performed but some important data are still missing. My comments are following.

  1. It seems that compounds prepared in the work is new. Please, add detailed procedure for their preparation because ref 25 deals with other complexes.
  2. Please, complete the characterization of these compounds by MALDI-TOF and NMR data. Add the values of molar extinction coefficients for each band.
  3. Please, precise if only one isomer of each compound was obtained or the complexes were preparzd as a mixture of isomers.
  4. Please, add the experimental procedure for preparation Co2Pc2(EP)4/PSS-Gr and Zn2Pc2(EP)4/PSS-Gr in the experimental section.
  5. Please, estimate size of nano-agregates of the complexes.
  6. Please, investigate the homogeneity of the distribution of iron atoms by EDAX mapping analysis and content of the metal by ICP analysis.
  7. Please, precise the images of which composite are present in Fig 2 (M = ?)
  8. Please, change the title of Fig 3 – three spectra are shown and the only one is mentioned in the title.
  9. Please, add a comparison of catalytic properties of novel composites with composites based on binuclear metallophthalocyanines previosly reported in the litterature.

Minor corrections:

  1. Line 21 : reaction(ORR). Please add a space.
  2. Line 156: p–p/transitions. Please, change to π-π
  3. Line 158: Please, delete one m.
  4. Line 161: 668nm to 741nm. Please, add two spaces.
  5. Line 539: N. S. Please, Delete space.
  6. Please check the abbreviation of journals in the references.

Author Response

Dear Editor Nora Pang,

We have studied the valuable comments from you and reviewers carefully, and tried our best to improve the manuscript and made some changes in the manuscript. These changes will not influence the content and framework of the paper. And here we did not list the changes but marked in red in revised paper. We appreciate for Editors/Reviewers’ warm work earnestly, and hope that the correction will meet with approval.

Responds to the reviewer’s comments

 Reviewer #1:

The manuscript by G. Zhang and W. Zhao and co-workers deals with the development of catalysts for oxygen reduction reaction. Novel binuclear metallophthalocyanines were prepared and deposited on GO. Then the composite materials were tested as catalysts in oxygen reduction reaction. Composite materials based on binuclear metallophthalocyanines were already studied in this reaction. In this work, the efficiency of their deposition on GO surface is explored. The work is carefully performed but some important data are still missing. My comments are following.

Comment 1It seems that compounds prepared in the work is new. Please, add detailed procedure for their preparation because ref 25 deals with other complexes.

Response: According to the reviewer’s comment, the detailed procedure has been added in the revised manuscript.

Preparation of M2Pc2(EP)4: Bisphthalonitrile (1.02 mmol), M(OAc)(0.52 mmol) and 4 mL DMAE were poured into a Teflon-lined autoclave at 220 ºC for 4h. The reaction mixture was then poured into methanol to produce a precipitate and the precipitate was washed sequentially with acetic acid, water and methanol. The crude product was then dissolved in DMF and reprecipitated by gradually adding methanol to the solution. The precipitate was washed again as in the previous method followed by centrifugation and dried at 100 ºC in an oven.

Comment 2Please, complete the characterization of these compounds by MALDI-TOF and NMR data. Add the values of molar extinction coefficients for each band.

Response: Thank you for your instructive suggestion. The mass spectra had been tested on a BRUKER MICROTOF-Q Ⅱ ESI-Q-TOF LC/MS/MS spectrometer. However, reflectron mode TOF-MS spectrum could not be obtained under the MALDI-MS conditions. Moreover, during the outbreak of coronavirus, the testing center was closed to the public, and the data of 1HNMR could not be supplemented within the required period. The structure of the M2Pc2(EP)4 compounds was studied by IR and UV-Vis spectra. To further illustrate the composition and chemical status of the as-prepared catalysts, the Fe2Pc2(EP)4 were also analyzed by XPS.

Comment 3Please, precise if only one isomer of each compound was obtained or the complexes were prepared as a mixture of isomers.

Response: Thank you for your careful work, the results of the characterization of these compounds indicated that each compound has only one isomer. This is consistent with the results in the literature[25,46]

Comment 4Please, add the experimental procedure for preparation Co2Pc2(EP)4/PSS-Gr and Zn2Pc2(EP)4/PSS-Gr in the experimental section.

Response: Thank you for your careful work, the experimental procedure for preparation Co2Pc2(EP)4/PSS-Gr and Zn2Pc2(EP)4/PSS-Gr in the experimental section has been added in the revised manuscript(Line 136-137).

Comment 5Please, estimate size of nano-agregates of the complexes. Please, investigate the homogeneity of the distribution of iron atoms by EDAX mapping analysis and content of the metal by ICP analysis.

Response: Thank you for your instructive suggestion. The structure of the M2Pc2(EP)4 compounds was studied by IR and UV-Vis spectra. To further illustrate the composition and chemical status of the as-prepared catalysts, the Fe2Pc2(EP)4 and Fe2Pc2(EP)4/PSS-Gr were also analyzed by XPS. Fig. 4a displays the XPS survey spectra of Fe2Pc2(EP)4 and Fe2Pc2(EP)4/PSS-Gr composites. The results showed that the Fe2Pc2(EP)4/PSS-Gr catalysts are composed of O, C, N and Fe elements, which further confirms the existence of Fe2Pc2(EP)4 in the as-prepared samples. The high-resolution Fe2p spectrum of the catalysts is shown in Fig.4b.

Figure 1. IR and UV-Vis spectra of M2Pc2(EP)4 compounds

Comment 6Please, precise the images of which composite are present in Fig 2 (M = ?)

Response: Thank you for your careful work. All of the images of composite are similar to each other, so we only showed the images of Fe2Pc2(EP)4/PSS-Gr composite(Figure 2).

Comment 7Please, change the title of Fig 3 – three spectra are shown and the only one is mentioned in the title.

Response: Thank you for your instructive suggestion. The title of Figure 3 had changed in revised manuscript;

Comment 8Please, add a comparison of catalytic properties of novel composites with composites based on binuclear metallophthalocyanines previosly reported in the litterature.

Response: According to the reviewer’s instructive suggestion, we have careful read some references. There have been many reports on the investigation and comparison of the electrocatalytic performances of various metallophthalocyanine-based materials on the ORR electrocatalysis in the literature. For instance, a study reported by Li and co-worker [1] concluded that the electrocatalytic performances of planar type binuclear Fe2Pc2 complexes are higher than those of mononuclear FePc ones. However, the performance of the planar type binuclear complex as Fe2Pc2/C (Vulcan XC-72) catalyst was significantly low in comparison with that of the Pt/C. In another report, Liu and his co-worker investigated the effect of graphene-based carbon materials on the electrocatalytic performance of a mononuclear FePc complex [2].

[1] T. Li, Y. Peng, K. Li, R. Zhang, L. Zheng, D. Xia, X. Zuo, Enhanced activity and stability of binuclear iron (III) phthalocyanine on graphene nanosheets for electrocatalytic oxygen reduction in acid, J. Power Sources 293 (2015) 511–518.

[2] D. Liu, Y.-T. Long, Superior Catalytic Activity of Electrochemically Reduced Graphene Oxide Supported Iron Phthalocyanines toward Oxygen Reduction Reaction, ACS Appl. Mater. Interfaces 7 (2015) 24063–24068.

Comment 9Minor corrections:

Line 21 : reaction(ORR). Please add a space.

Line 156: p–p/transitions. Please, change to π-π

Line 158: Please, delete one m.

Line 161: 668nm to 741nm. Please, add two spaces.

Line 539: N. S. Please, Delete space.

Response: We are so sorry for the mistakes, we have checked and modified the mistakes in the revised manuscript.

Comment 10Please check the abbreviation of journals in the references.

Response: We are so sorry for the mistakes, we have checked the abbreviation of journals in the references and modified the mistakes in the revised manuscript.

Reviewer 2 Report

The paper of Zhang and co-workers describes the application of the phtalocyanine complexes in the oxygen reduction reaction. While I believe there is potential in this work and I guess the scientific procedure is sound, there is significant improvement needed with respect to the quality of the presentation. The paper is written in a very chaotic manner and it more resembles the draft thatn the completed manuscript. It is extremely difficult to follow and thus to judge its quality. Some of more specific pints:

1/ The difficulty to understand what the authors mean starts with the title: "Metallophthalocyanines Graphene-based Oxygen Reduction Reaction Catalysts". Is the oxygen reduction reaction based on graphene? or the catalysts are metalophtalocyanine *and* graphene based?

2/ The figures are of very low resolution and the numbers put there are illegible - for instance Figure 3: I can barely see the values in the x axis; Figure 4: the same and additionally the peaks of the curves are in bold which makes them even less legible.

3/ The caption of Figure 1 is overly simplified: what do the pink spheres represent? the fully formed M2Pc2 complex, or only the linkers between the macrocycles? Similarly in Figure 2: what does the white ring and the arrow represent? This information should be given in the caption without needing to look in the text.

4/ Section 2.3.2. describes the preparation of Fe2Pc2(EP)4/PSS-Gr, but there are no respective sections on the preparation of systems with Co and Zn. Also: what does the EP stand for?

5/ The discussion in the section 3.2 (which is again poorly worded) is not convincing. t2g and eg states are occupied differently depending on the coordination and the spin state of the system. Besides obvious 4 coordination sites provided by the macrocycle, there is no evidence on the axial coordination.

6/ Conclusion: "Moreover, the pi-pi supramolecular interactions between the
M2Pc2(EP)4 and PSS-Gr dramatically enhanced the pi electron clouds in conjugated structure and oxygen could be reduced much easily." which result actually supports this claim?

7/ Other minor issues: full-filled -> fully filled or fully occupied, substitutes -> substituents

Author Response

Dear Editor Nora Pang,

We have studied the valuable comments from you and reviewers carefully, and tried our best to improve the manuscript and made some changes in the manuscript. These changes will not influence the content and framework of the paper. And here we did not list the changes but marked in red in revised paper. We appreciate for Editors/Reviewers’ warm work earnestly, and hope that the correction will meet with approval.

Responds to the reviewer’s comments

 Reviewer #2:

The paper of Zhang and co-workers describes the application of the phtalocyanine complexes in the oxygen reduction reaction. While I believe there is potential in this work and I guess the scientific procedure is sound, there is significant improvement needed with respect to the quality of the presentation. The paper is written in a very chaotic manner and it more resembles the draft thatn the completed manuscript. It is extremely difficult to follow and thus to judge its quality. Some of more specific pints:

Comment 1The difficulty to understand what the authors mean starts with the title: "Metallophthalocyanines Graphene-based Oxygen Reduction Reaction Catalysts". Is the oxygen reduction reaction based on graphene? or the catalysts are metalophtalocyanine *and* graphene based?

Response: Thank you for your careful work. The oxygen reduction reaction was based on graphene and metalophtalocyanine. The electrons on PSS-Gr migrate to Fe2Pc2(EP)4 and PSS-Gr enhanced the electrocatalytic activity of Fe2Pc2(EP)4.

Comment 2The figures are of very low resolution and the numbers put there are illegible - for instance Figure 3: I can barely see the values in the x axis; Figure 4: the same and additionally the peaks of the curves are in bold which makes them even less legible.

Response: Thank you for your careful work. All the pictures in my manuscript are clear and normal. Maybe there's something wrong with the file format conversion. We corrected the pictures in our revised manuscript (Shown in Figure 3, Figure 4).

Comment 3: The caption of Figure 1 is overly simplified: what do the pink spheres represent? the fully formed M2Pc2 complex, or only the linkers between the macrocycles? Similarly in Figure 2: what does the white ring and the arrow represent? This information should be given in the caption without needing to look in the text.

Response: Thank you for your careful work. We had corrected the information in revised manuscript (Shown in Figure 1, Figure 2).

Comment 4: Section 2.3.2. describes the preparation of Fe2Pc2(EP)4/PSS-Gr, but there are no respective sections on the preparation of systems with Co and Zn. Also: what does the EP stand for?

Response: Thank you for your careful work, the experimental procedure for preparation Co2Pc2(EP)4/PSS-Gr and Zn2Pc2(EP)4/PSS-Gr in the experimental section has been added in the revised manuscript.

Comment 5: The discussion in the section 3.2 (which is again poorly worded) is not convincing. t2g and eg states are occupied differently depending on the coordination and the spin state of the system. Besides obvious 4 coordination sites provided by the macrocycle, there is no evidence on the axial coordination.

Response: Thank you for your careful work, a mechanism was proposed for ORR catalyzed by Fe2Pc2(EP)4/PSS-Gr based on experiment results and references. Bridge adsorption can coordinate one oxygen molecule with two central metal ion Fe(II), so it can form more stable peroxide intermediates, which is more conducive to oxygen-oxygen bond(O-O) breaking. In a critical paper reported by Tanaka et al[3] on the electrocatalytic activity of mononuclear MPcs in ORR, most of the findings on the FePc-dioxygen interactions points out the formation of Fe(III)-O2- species. The quantum mechanical considerations also suggest that the electronic configuration of Fe(III)-O2- species is an optimum state for the subsequent activation and reduction of dioxygen. In this work, the involvement of cofacial dual central metals has a great importance for the efficiency of the M2Pc2 structure as the electrocatalyst for ORR. It can be estimated that the oxidation of not only dual but also cofacial Fe(II) centers in the M2Pc2-type dimeric structure leads to the formation of Fe(III)-O-O-Fe(III) peroxo species.

[1]Communication—High-Performance and Non-Precious Bifunctional Oxygen Electrocatalysis with Binuclear Ball-Type Phthalocyanine Based Complexes for Zinc-Air Batteries. Journal of The Electrochemical Society, 2016, 163 (9) A2001-A2003;

[2] Electrocatalytic Activity of Novel Ball-Type Metallophthalocyanines with Trifluoro Methyl Linkages in Oxygen Reduction Reaction and Application as Zn-Air Battery Cathode Catalyst. Electrochim. Acta. 2017, 233, 237–248;

[3] Electrocatalytic aspects of iron phthalocyanine and its. mu.-oxo derivatives dispersed on high surface area carbon. J. Phys. Chem. 1987, 91, 3799–3807.

[4] Electrochemical Effects of Lithium-Thionyl Chloride Battery by Central Metal Ions of Phthalocyanines-Tetraacetamide Complexes. J. Electrochem. Soc. 2017, 164,A3628-A3632

Comment 6: Conclusion: "Moreover, the pi-pi supramolecular interactions between the M2Pc2(EP)4 and PSS-Gr dramatically enhanced the pi electron clouds in conjugated structure and oxygen could be reduced much easily." which result actually supports this claim?

Response: Thank you for your careful work, the π-π supramolecular interactions between the M2Pc2(EP)4 and PSS-Gr enhanced the electrocatalytic activity. The electrons on PSS-Gr migrate to Fe2Pc2(EP)4 and PSS-Gr enhanced the electrocatalytic activity of Fe2Pc2(EP)4.

Comment 7: Other minor issues: full-filled -> fully filled or fully occupied, substitutes -> substituents

Response: We are so sorry for the mistakes, we have checked the whole manuscript and modified the mistakes in the revised manuscript, we had corrected the information in revised manuscript.(Shown in Line17, 74, 217, 225-226)

Round 2

Reviewer 1 Report

Recommendation: The manuscript is publishable after a minor revision.

  1. Zhang and W. Zhao and co-workers try to publish the manuscript without any additional experimental work on the characterization of compounds and materials.

The data presented by the authors do not allow to conclude on the isomer purity of the obtained compounds as well as on the purity of the studied compounds. The references 25 and 46 do not give the isomer composition and they also deal with different compounds. The homogeneity of material can be proved only by EDAX mapping analysis.

Let’s give the authors a possibility to discuss these questions with the scientific audience of Nanomaterials.

However, the author should add in the body of the article a detailed comparison of catalytic properties of novel composites with composites based on metallophthalocyanines previously reported in the literature to show the value of their work.

Please, note that COVID is already responsible for many human problems. May be there is no reason to do it responsible for any human problem.

Author Response

Dear reviewer,

    We’re so sorry for taking the coronavirus as the reason. We believe that human beings will defeat the new coronavirus before long. Thank you for your instructive suggestion. We have studied the valuable comments from you carefully, and tried our best to improve the manuscript and made some changes in the manuscript. These changes will not influence the content and framework of the paper. And here we did not list the changes but marked in red in revised paper. We appreciate for your warm work earnestly, and hope that the correction will meet with approval.

    The results of reports pointed out the possibility of the increas of the catalytic performance by the modification of the main M2Pc2 skeleton and encouraged us to design new phthalocyanine-based catalysts and test their performances. The performance of M2Pc2(EP)4 with methoxy linkages was lower than that of the metallophthalocyanines with trifluoro methyl linkages [25]. The electrophilicity of trifluoro methyl groups is higher than that of the methoxy groups. The highly electrophilicity of trifluoromethyl groups on the macrocycles dramatically enhances the electrocatalytic performance. The comparison of the performances of M2Pc2(EP)4 catalysts used in this work and in reported article points out the importance of the bridging units in the ball-type structure.(Line 60-65)

     More significantly, PSS-Gr dramatically enhanced the electrocatalytic activity of M2Pc2(EP)4. PSS-Gr provide pathway for fast electron transferring and prevent the aggregation of M2Pc2(EP)4 catalysts. M2Pc2(EP)4 were loaded on the surface of PSS-Gr to enhance the catalytic activity and stability for ORR based on the π-π supramolecular interaction between MPc molecules and graphene. The results indicated that the catalytic performance of M2Pc2Rn could be enhanced by the modification of π electron conjugated structure of M2Pc2(EP)4 and carbon materials. We’ll design more phthalocyanine-based catalysts and test their performances in our continuing studies.(Line 298-308)

Reviewer 2 Report

The paper has not changed much since my initial review, in which I requested a major revision - and therefore I am not changing my recommendation. My most important objection that the paper is poorly written still holds, and with the changes of the pictures it is now IMPOSSIBLE to follow. What am I supposed to understand from this:

My advise to the authors is: please rewrite the manuscript, without highlighting any changes, because the changes needed are extensive. Please do not use extensive formatting and make sure the quality is acceptable. Please have the manuscript corrected by a linguist. Otherwise it is just a waste of time.

Author Response

Dear reviewer,

    We are so sorry for our unprofessional attitude; According to the reviewer’s instructive suggestion, we have tried our best to revise our manuscript and use a professional English editing service according to the comments. Attached please find the revised version, which we would like to submit for your kind consideration.

Round 3

Reviewer 2 Report

This is a prime example of how important is the quality of the presentation. I am happy to recommend the revised version for publication in Nanomaterials.